# Differential Effects of the Processed and Unprocessed Garlic (*Allium sativum* L.) Ethanol Extracts on Neuritogenesis and Synaptogenesis in Rat Primary Hippocampal Neurons

**DOI:** 10.3390/ijms241713386

**Published:** 2023-08-29

**Authors:** Yeasmin Akter Munni, Raju Dash, Ho Jin Choi, Sarmistha Mitra, Md. Abdul Hannan, Kishor Mazumder, Binod Timalsina, Il Soo Moon

**Affiliations:** 1Department of Anatomy, College of Medicine, Dongguk University, Gyeongju 38066, Republic of Korea; yeasminakteracce@gmail.com (Y.A.M.); rajudash.bgctub@gmail.com (R.D.); chjack@naver.com (H.J.C.); sarmisthacu@gmail.com (S.M.); hannanbmb@bau.edu.bd (M.A.H.); binodtimalsina19@gmail.com (B.T.); 2Department of New Biology, Daegu Gyeongbuk Institute of Science and Technology, Daegu 42988, Republic of Korea; 3Department of Biochemistry and Molecular Biology, Bangladesh Agricultural University, Mymensingh 2202, Bangladesh; 4Department of Pharmacy, Jashore University of Science and Technology, Jashore 7408, Bangladesh; kmazumder@just.edu.bd; 5School of Optometry and Vision Science, UNSW Medicine, University of New South Wales (UNSW), Sydney, NSW 2052, Australia

**Keywords:** *Allium sativum* L., black garlic, linalool, neuritogenesis, synaptogenesis, GC-MS

## Abstract

Garlic (*Allium sativum* L.) is an aromatic herb known for its culinary and medicinal uses for centuries. Both unprocessed (white) and processed (black) garlic are known to protect against the pathobiology of neurological disorders such as Alzheimer’s disease (AD) and Parkinson’s disease (PD), which has been attributed to their anti-inflammatory and antioxidant properties. The information on the effects of processed and unprocessed garlic on neuronal process outgrowth, maturation, and synaptic development is limited. This study aimed at investigating and comparing the effects of the ethanol extracts of unprocessed (white garlic extract, WGE) and processed (black garlic extract, BGE) garlic on the maturation of primary hippocampal neurons. Neurite outgrowth was stimulated in a dose-dependent manner by both WGE and BGE and the most effective doses were 15 μg/mL and 60 μg/mL, respectively, without showing cytotoxicity. At this optimal concentration, both extracts promoted axonal and dendritic growth and maturation. Furthermore, both extracts substantially increased the formation of functional synapses. However, the effect of WGE was more robust at every developmental stage of neurons. In addition, the gas chromatography and mass spectrometry (GC-MS) analysis revealed a chemical profile of various bioactives in both BGE and WGE. Linalool, a compound that was found in both extracts, has shown neurite outgrowth-promoting activity in neuronal cultures, suggesting that the neurotrophic activity of garlic extracts is attributed, at least in part, to this compound. By using network pharmacology, linalool’s role in neuronal development can also be observed through its modulatory effect on the signaling molecules of neurotrophic signaling pathways such as glycogen synthase kinase 3 (GSK3β), extracellular signal-regulated protein kinase (Erk1/2), which was further verified by immunocytochemistry. Overall, these findings provide information on the molecular mechanism of processed and unprocessed garlic for neuronal growth, survival, and memory function which may have the potential for the prevention of several neurological disorders.

## 1. Introduction

Neurodegenerative diseases (NDDs) are characterized by the dysregulation of neurogenesis, which in turn impacts neuronal differentiation, global or regional loss of neurons, and compromised neural network accompanying cognitive deficits [1,2]. Neurotrophic factors (NTF) that play a crucial role in promoting neuronal function, survival, and maturation are critical in the pathobiology of NDDs [3,4]. An insufficient level of various NTFs, including brain-derived neurotrophic factor has been reported in NDDs [5,6], thereby, NTFs or their mimetic are of therapeutic value. Evidence suggests that NTFs mimetic can stimulate neuritogenesis and help reconstruct compromised neuronal circuits [7], thus holding promise in the development of therapeutic agents for NDDs. Small molecule NTF mimetics have benefits over NTFs due to their accessibility to the blood–brain barrier [7]. As NTF mimetics, various medicinal herbs and their bioactive phytochemicals have emerged as a promising source of novel therapeutic agents for the treatment of neurodegenerative illnesses in recent years where the neurotrophic effects of these natural products have been linked to an increase in neuronal outgrowth [8,9,10,11]. Focusing on the neurotrophic effects of the bioactive components, the aim of this present study is to investigate the progression of neurons from the nascent neuronal differentiation stage to the synapse formation of the ethanol extract of unprocessed and processed *A. sativum* L.

The bulbous flowering plant known as garlic (*Allium sativum* L.), which belongs to the genus Allium and family Liliaceae, has been used as a spice or seasoning for thousands of years throughout South Asia, Central Asia, and northeastern Iran [12]. Garlic is considered an important ingredient in the Ayurveda formula because of its preventive properties in cardiovascular diseases, its ability to regulate blood pressure, blood sugar, and cholesterol levels, its effectiveness against bacterial, viral, and fungal infections, its ability to enhance the immune system, and its antitumor, anti-inflammatory, and antioxidant properties [13,14]. Garlic has a rich phytochemical profile, including phenolic acids, steroids, flavonoids, and anthocyanins [15]. Garlic and its secondary metabolites have strong neuroprotective effects with memory enhancement capacities. Recent studies have shed light on the neuroprotective effects of allicin and S-allyl-cysteines (SAC), making it an intriguing potential treatment option for AD [16,17,18]. The positive effects of aged garlic extract and its components on memory and neuroprotection in neurodegeneration have also been reported in numerous studies dating back to the 1990s [19,20,21]. However, there have been no reports on the effects of processed and unprocessed garlic (*A. sativum*) on neuritogenesis and synaptogenesis.

In this study, we compared synaptogenesis and cytoarchitecture complexity and the early neuronal maturation of BGE and WGE. In addition, to understand the chemical content and quality of plant extracts, GC-MS analysis was carried out, revealing several phytochemicals, including linalool being common in both extracts. We further demonstrated the involvement of bioactive metabolite linalool in the stimulation of neuritogenesis. Moreover, the network pharmacology technique was subsequently implemented to gain mechanistic insight into the bioactive component, and the potential target was confirmed by immunocytochemistry. Together, this combined study of in vitro and network pharmacology showed that BGE and WGE promote neuronal differentiation, axodendritic outgrowth, and synaptic development.

## 2. Results

### 2.1. Dose Optimization of Garlic Ethanol Extract for Neurite Outgrowth Activity

Hippocampal neurons of embryonic-19 (E19) rats were treated with WGE and BGE or a vehicle at varying doses. After 3 days, images of the neurons were obtained. The phase-contrast micrographs were used to investigate the morphology of 30 different neural cells on DIV3 (Days in vitro 3) for the dose optimization of BGE and WGE for neurite outgrowth activity (Figure 1). The typical fluorescence images of WGE and BGE-treated neurons were shown in Figure 2A, where both extracts promoted neuronal development dose-dependently (Appendix A). The results demonstrated that neurite outgrowth parameters such as total neurites numbers and length, and the length of the longest process were significantly increased in both WGE (Figure 2(Ba)) and BGE (Figure 2(Bb))-treated neurons compared to control at a concentration of 15 and 60 μg/mL, respectively. Based on these findings, these optimal concentrations were chosen for further experiments. In addition, we compared the neuritogenic activity of BGE and WGE to that of other known neuromodulatory substances (positive control) such as puerarin (PE) and scoparone. Neurons were fixed at DIV3 and immune-stained for α-tubulin to observe hippocampal neuron morphologies (Appendix A). It was found that both PE and scoparone promoted the growth of neurites such as the number and length of the total primary neurites as well as the longest length of the primary neurite, although not as much as BGE and WGE (Appendix A).

### 2.2. Effects of BGE and WGE on Neuronal Viability

The effects of WGE and BGE on neuronal viability were evaluated using trypan blue dye exclusion assay on DIV8. At the indicated concentrations (7.5–75 μg/mL), both BGE (Appendix A) and WGE (Appendix A) did not show cytotoxicity, but rather slightly promoted viability when compared to the control culture (cell viability ~90%). Therefore, BGE and WGE were found to be relatively safe for neurons at the indicated doses.

### 2.3. Early Neuritogenic Activity of BGE and WGE

We observed the effect of BGE and WGE on neuronal polarity at the very early developmental stage (Figure 1). Staining with anti-MAP2 (green) as a somatodendritic marker and anti-Tau (red) as an axonal marker was carried out to differentiate various phases of early neuronal development of WGE and BGE-treated hippocampal cultures compared with vehicle (Figure 3(Aa,Ba)). Cultures treated with WGE and BGE showed approximately 15% and 10% more cells, respectively, showing the extension of axons, in the developmental stage 3 category than the vehicle-treated neuron at 24 h (DIV1) (Figure 3(Ab)). At 48 h (DIV2), axonal elongation of WGE and BGE-treated neurons at stage 3 was observed in around 55% and 50%, respectively, of total neurons, whereas only 16% of total neurons treated with vehicle reached stage 3 (Figure 3(Bb)). These findings suggest that WGE and BGE improved neural differentiation, while WGE impact is more pronounced.

### 2.4. Effect of BGE and WGE on Axonal Development

Next, we investigated the effect of BGE and WGE on the development of axonal sprouting. We used a morphometric study to determine whether WGE and BGE have an impact on axonal maturation. Within five days of treatment with both extracts, neurons had undergone enormous axonal growth. Immuno-staining with ankyrin G was used to differentiate axons from dendrites (Figure 4A). The axonal length of cells treated with BGE and WGE was found to increase significantly over time, by 31.25% and 56.25% relative to control (Figure 4B). More complex axon collateral branches were observed in hippocampal neurons grown in WGE-treated cultures compared to vehicle controls. The numbers of primary and secondary branches of WGE-treated neurons were increased by 51.51% and 87.5% over the control (Figure 4C). The primary and secondary collateral branches in BGE and WGE-treated neurons increased 2-fold and 2.5-fold longer compared to the control (Figure 4D). However, the number of tertiary branches was found to rise dramatically after WGE treatment, but the other parameter (length) remained unaffected (Figure 4C,D).

In addition, we performed a Sholl analysis where a significant increase in the number of axonal intersections was seen in the neurons treated with WGE and BGE (Figure 4E). The axonal intersections and branch points in the Sholl circles reached up to a maximum of 220 and 200 µm, respectively, in WGE-treated neurons while vehicle-treated neurons only reached 160 and 150 µm, respectively. On the other hand, in BGE-treated neurons, there was no axonal intersection or branch point that could be seen beyond the spheres of 190 and 180 µm, respectively (Figure 4F,G).

### 2.5. Effect of BGE and WGE on Dendritic Morphological Complexity

Stage 4 of neuronal development, or dendritic elongation and tree-like branching, typically happens 3–7 days after neuronal cultures. Therefore, the effect of WGE and BGE on dendritic arborization was investigated during the earliest phases of development (DIV5). The number of primary dendrites increased after the treatment with BGE and WGE (nearly 30.76% and 53.84% of control, respectively) (Figure 5A), as well as the total length of primary dendrites (nearly 13.95% and 39.53% of control, respectively) (Figure 5B). We also assessed the total number and length of the dendritic branches (both main and secondary) and found that WGE-treated neurons had significantly higher values than BGE-treated neurons in comparison to the control (Figure 5C,D), although both BGE and WGE-treated cells exhibited the same effect on the number of secondary branches. 

Sholl analysis, which quantifies dendritic projections by counting their intersections at various radial distances from the cell body, was also used, which gives the most efficient route for synaptic currents. As shown in Figure 5E, dendritic intersections in neurons treated with BGE and WGE reached up to 100 µm circles, whereas dendritic intersections in neurons treated with vehicles reached beyond 70 µm circles. However, in BGE-treated neurons, branching points were observed up to 80 µm from the center of soma, whereas WGE reached up to 70 µm, and in control neurons, no branching point was noted beyond 50 µm (Figure 5F).

### 2.6. Effect of BGE and WGE on the Synaptic Formation

To investigate the effect of processed and unprocessed garlic on synaptogenesis, we carried out an immunoblotting analysis and observed the protein expression of a synaptic molecular marker PSD95 (postsynaptic density-95). As PSD95 is a known remarkably stable protein in PSDs at excitatory synapses, in order to determine whether BGE and WGE enhanced the expression of PSD95, we incubated the membrane with an anti-PSD95 antibody. Figure 6(Aa) represents the higher expression of PSD95 in BGE and WGE-treated neurons for DIV16. Here, alpha-tubulin was used as a loading control. The mean intensity of the PSD95 band was quantified and both extracts showed a significant increase in PSD95 expression compared to the control (Figure 6(Ab)). Further, we conducted immunocytochemistry (ICC) to validate the western blot data where neurons were immune-stained with presynaptic marker, SV2 (Synaptic vesicle 2), and postsynaptic marker PSD95 (Figure 6(Ac)). Quantification of the amount of SV2-immunoreactive, PSD95-immunoreactive, and their co-localized puncta demonstrated that there was a considerable increase in the development of synapses in both extract-treated neurons, where BGE showed a slightly higher number of puncta than WGE (Figure 6(Ac)).

The two most prevalent N-methyl-D-aspartate receptor (NMDAR) subtypes, GluN2A/NR2A and GluN2B/NR2B receptors, have been shown to play important roles in both synaptic and extra-synaptic processes, including memory and cognition. Thus, we utilized an immunoblotting experiment to explore the effect of both extracts on NMDAR-mediated synapse formation (Figure 6(Ba)). The expressions of both NR2A and NR2B were significantly increased in neurons that were treated with WGE and BASE, but the level of NR2A protein expression was higher in the BGE-treated neurons (Figure 6(Bb)). Moreover, immunostaining with NR2A and NR2B against the presynaptic marker SV2 was used to determine whether both extracts influenced NMDAR-mediated synaptogenesis (Figure 6(Ca,Cb)). BGE (~2.1 fold) and WGE (~1.6 fold) increased the number of NR2A (Figure 6(Cc)), while NR2B puncta for BGE and WGE-treated neurons were increased by ~1.94 fold and ~1.97 fold, respectively, as compared with controls (Figure 6(Cd)). Overall, these findings imply that both processed and unprocessed garlic have potential effects on NMDAR-mediated synaptogenesis.

### 2.7. Phytochemical Profiling of Garlic Extracts by GC/MS Analysis

GC-MS analysis revealed a total of 24 compounds in processed garlic (BGE) (Appendix A) and 21 compounds in unprocessed garlic (WGE) (Appendix A). Of these, linalool and trifluoromethyl T-butyl sulfide were common metabolites in both extracts. GC-MS analysis of processed garlic showed 14.59833% of the peak area with a retention time of 9.17 min for linalool. In contrast, linalool in unprocessed garlic was represented by a peak area of 0.71% with the same retention time. Being one of the components of garlic extracts, linalool was added to the neuronal culture to investigate whether this compound could stimulate neuronal growth as an active component of garlic that is responsible for its neuronal outgrowth activity.

### 2.8. Linalool Stimulates Neurite Outgrowth in Hippocampal Neurons

Linalool treatment (10–70 nM) resulted in significantly longer neurites on average with a higher number of primary neurites as compared to vehicle-treated neurons (Figure 7A,B), indicating that linalool might contribute, at least in part, to the neurite outgrowth activity of garlic extracts. Nonetheless, the impact was significantly lower than BGE and WGE garlic, suggesting that some other phytochemicals might also play an important role.

### 2.9. Network Pharmacology Analysis

To find out the toxicity and bioavailability of linalool, we performed absorption, distribution, metabolism, and excretion/transport (ADME/T) analysis. The #stars score was less than 4 (ranging from 0–5) while the brain/blood partition coefficient (QPlogBB) was found at 0.015 (−3.0–1.2, recommended range). Linalool satisfied the rules of five by maintaining the mol_MW < 500, donor HB ≤ 5, accptHB ≤ 10 and QPlogPo/w < 5 (Appendix A).

A total of 309 genes were identified as linalool’s potential targets. Likewise, by using the Gene Cards dataset, 384 target genes associated with neuronal development (ND) were collected (Appendix A). An overlap of 23 target genes was found as linalool-ND common targets, which was shown in Figure 8A through a Venn diagram. The protein–protein interaction (PPI) network was generated with Cytoscape v3.9.0 software (Figure 8B) following String databases where 10 hub genes were represented by green nodes.

Gene Ontology (GO) enrichment analysis showed that the biological process (BP) term was mostly involved in response to oxidative stress, cellular response to growth factor stimulus, and response to growth factor (Figure 8C). Postsynapse, synapse, microtubule skeleton, cell leading edge, and microtubule organizing center were seen in the cellular component (CC) term (Figure 8D). On the other hand, the molecular function (MF) term was associated with the phosphatase binding, protein serine/threonine kinase activity, transcription factor binding, and enzyme regulator activity (Figure 8E). The top 20 most important signaling pathways were selected for Kyoto Encyclopedia of Genes and Genomes (KEGG) enrichment pathway analysis (Figure 8F). Among the various cell signaling pathways, the neurotrophin signaling pathway and mitogen-activated protein kinase (MAPK) signaling pathway were observed that are essential for neuronal survival, neurite formation, and development, which includes the formation of synapses and synaptic plasticity. Additionally, the KEGG pathway mapper (Appendix A) was also analyzed, and it showed five major targets involved in neuronal development. These targets are GSK3β, c-Jun (Jun proto-oncogene AP-1 transcription factor subunit), Erk1/2, Akt (AKT serine/threonine kinase 1), and P53 (tumor protein p53) which might be potentially modulated by the bioactive component of BGE and WGE.

### 2.10. Validation of Target Genes by Immunocytochemistry

Using ICC, we evaluated the expression of the genes during neural differentiation (p-Erk1/2, p-GSK3β) to confirm the modulation of distinct genes by linalool as found in the network pharmacology (Appendix A). The relative fluorescence intensity of Erk1/2 were significantly increased (Appendix A) while p-GSK3β expression was decreased in linalool and BGE and WGE-treated hippocampal neurons (Appendix A), indicating the differential role of linalool and the parent extracts in neurite development in primary hippocampal neurons.

## 3. Discussion

Neurodegeneration is a common pathological phenomenon in age-associated neurological disorders, including AD and PD. Natural products with the ability to potentiate neuritogenesis and synaptogenesis, and to protect against neuronal damage are of therapeutic significance in the prevention and treatment of these neurodegenerative disorders (NDDs). In this cell-based in vitro experimental approach, we demonstrated that processed (BGE) and unprocessed (WGE) garlic extracts induced neuronal differentiation and promoted growth and maturation of axons, dendrites, and synapses in hippocampus neurons, indicating that garlic extracts play a significant role in every stage of neuronal development and suggests its pharmacological value in the management of NDDs. 

Neurons start as simple spheres, but throughout the course of their lives, they can develop into more complex structures that have a multitude of dendrites that are heavily arborized with lengthy axons [22,23]. We observed that BGE and WGE exerted their neurotrophic effects throughout the entire process of all three stages including (lamellipodia, stage 1), neurites begin to expand from the cell body, (minor processes, stage 2) rapid outgrowth of neurites into the axon and polarization of the cell (axon formation, stage 3) [22,24]. Axonal maturation is thought to be responsible for the formation of primary brain communication and is a potential therapeutic target for neurodegenerative diseases [25,26]. Furthermore, WGE and BGE were able to promote both axonal sprouting. However, the effect of WGE was more pronounced in the early stages of neuronal differentiation, along with axonal and dendritic arborization. Moreover, the viability assay demonstrated that both extracts increased the survival rate at all concentrations, suggesting that garlic extracts could be viable sources of phytochemicals that can sustain healthy neurons.

Synaptogenesis is an essential event for the neuronal growth and development associated with learning and memory [27]. The machinery for neurotransmitter release can be modified both presynaptically and postsynaptically, and the receptor composition and activity can alter the strength of synapses [28]. In this study, higher levels of the presynaptic marker, SV2, and postsynaptic marker, PSD95, along with the synapses (co-localized puncta), were observed in BGE and WGE-treated neurons, suggesting the potential role of garlic extracts in influencing synaptogenesis. Similarly, the levels of two subunits of excitatory NMDARs, GluN2A (NR2A), and GluN2B (NR2B), were shown to be considerably higher in BGE-treated neurons compared to WGE-treated neurons. Since NMDARs are crucial in synaptic plasticity, learning, and memory [29,30], the current role of garlic extracts in the promotion of synaptogenesis and modulation of synaptic receptor proteins suggests their significance in the therapeutic management of NDDs that have compromised network connectivity.

To explore the bioactive secondary metabolites of garlic that might be responsible for its neurotrophic activity, we employed GC-MS analysis of both unprocessed and processed garlic extracts and found that both extracts contained diverse types of compounds. Of these, linalool and trifluoromethyl T-butyl sulfide were common metabolites of both WGE and BGE. Linalool has been reported to exert a wide range of pharmacological actions including antibacterial, anti-inflammatory, anti-cancer, and antioxidant properties [31]. Linalool also plays an important role in the reduction of Aβ amyloid neurotoxicity by suppressing the free radical generation in the rat hippocampus [32]. Another study reported the neuroprotective effect of linalool from the damage caused by oxygen–glucose deprivation-induced neuronal injury [33]. Although linalool was reported to show a neuroprotective effect, this is the first study that reported the neuritogenic activity of linalool in primary cultured neurons. Although not revealed by GC-MS analysis, garlic also contains several organosulfur compounds that have neuroprotective and neurotrophic activity [34]. In addition to linalool, garlic extract also contains other phytochemicals, such as dodecanoic acid (lauric acid) that may also contribute to the neurotrophic activity, as lauric acid can protect neurons from oxidative stress and inflammatory cytokines, and improve behavioral impairments in HPD (haloperidol-induced PD) rats [35]. Moreover, garlic and its bioactive metabolites are well-known for their antioxidant, anti-inflammatory, and immunomodulatory activities that further contribute to neuronal survival in adverse conditions [19,36]. The current findings along with the previous evidence suggest that both BGE and WGE could be a potential source of neurotrophic factor mimetics that can modulate molecular pathways associated with growth and maturation of neurons and functional connectivity.

Numerous signaling pathways and proteins are responsible for promoting neuronal differentiation and development. Using the network pharmacology approach, we demonstrated the top ten genes with high degree scores from PPI, identifying several genes implicated in neurotrophic activity including GSK3β, c-Jun, Erk1/2, Akt, and P53 by KEGG pathway enrichment analysis. p-GSK3β and p-Erk1/2 were found to be regulated in the linalool, BGE, and WGE-treated neurons. For instance, emerging evidence suggests that GSK3β is a significant modulator of neuronal development such as neuronal polarization, axon development, and axonal sprouting [37], while GSK3β activity was decreased by phosphorylation at its Ser-9 [38,39]. Both BGE and WGE and the bioactive compound, linalool-treated neurons, reduced the expression of p-GSK3β compared to the vehicle, indicating the GSK3β-promoting activity in neuronal development. In addition, BGE, WGE, and their active metabolite, linalool, increased the expression of p-Erk1/2. MEK1/2 (MAPK) is an upstream activator kinase that is known to activate Erk1/2 by phosphorylating threonine and tyrosine residues, which in turn regulates the transcription of several genes involved in neurite outgrowth and axonal, dendritic sprouting, synaptogenesis, and memory formation [40,41,42]. Therefore, BGE and WGE might upregulate Erk1/2 and GSK3β signaling pathways to promote neuritogenesis and synaptogenesis.

Overall, the combined studies from in vitro and network pharmacology approaches suggest the mechanistic understanding of processed and unprocessed garlic on neuronal differentiation and synaptogenesis.

## 4. Materials and Methods

### 4.1. Chemicals and Reagents

Linalool (purity ≥ 98%) was purchased from Medchemexpress (Cat No: HY-N0368; Monmouth Junction, NJ, USA) and dissolved in dimethyl sulfoxide (DMSO) with a stock concentration of 10 nM. Scoparone (purity ≥ 98%) was obtained from Cayman Chemical (Cat No: 120-08-1; Ann Arbor, MI, USA) and dissolved in ethanol, yielding a 60 mM stock solution. At a concentration of 20 mM, puerarin (purity ≥ 98%) was originally acquired from Sigma-Aldrich (Cat No: 3681-99-0; St. Louis, MO, USA) and was dissolved in ethanol (EtOH). After being produced, all of these compounds were chilled to a temperature of −20 °C for further use. Medium and supplements for cell culture were bought from Invitrogen in Carlsbad, CA, USA unless stated otherwise.

### 4.2. Collection and Extract Preparation of WGE and BGE

The fresh garlic (*A. sativum* L.) was purchased at a market in the southeastern portion of North Gyeongsang Province in South Korea and the botanical name was verified by an expert. The plant name has been checked with http://www.theplantlist.org, accessed on 30 October 2022. Garlic cloves were rinsed with fresh water. Peels were removed, sliced, and dried completely at room temperature (RT). A powdered garlic sample was prepared by using a Pottery Mortar set. For the extraction, two grams of garlic powder were soaked into 95% ethanol (1:50 *w*/*v*) and agitated at 200 rpm (Shaking incubator; VS-8480) for 48 h at RT. After shaking, the extract was filtered using Advantec No. 2 filter paper (Advantec MFS, Inc., Sierra Trinity, CA, USA) and dried completely. An aliquot (8 mg/mL) of the unprocessed garlic extract was prepared in Dimethyl sulfoxide (DMSO) and kept at −20 °C in a vial that was wrapped in foil for further experiments.

To prepare the processed garlic extract, fresh garlic was allowed to undergo a process of natural aging in a rice cooker at a temperature of 140 or 150 °F and a humidity level of 75 percent for a period of 20 to 25 days [43]. After that, garlic bulbs were removed from the rice cooker allowing them to cool at RT. After that, the dried cloves were pulverized into fine powder. Extraction was performed following the same procedure as for white garlic extract preparation. The dried extract was reconstituted to make an aliquot (16 mg/mL) in DMSO. A voucher specimen for both fresh garlic and processed garlic extracts has been deposited at Il Soo Moon’s Lab (Dongguk University College of Medicine, Department of Anatomy).

### 4.3. GC-MS Analysis

Gas chromatography and mass spectrometry (GC-MS) were used to evaluate the chemical composition and active components of *A. sativum* L. extract (processed and unprocessed). The GC-MS analysis of *A. sativum* extract was performed by using a 7890A capillary gas chromatographic system (Agilent Technologies, Santa Clara, CA, USA) organized with a mass spectrometer of 95% dimethyl-poly-siloxane and 5% phenyl. The size of the Silica capillary column (HP-5MSI, film: 0.25 µm) was 0.25 mm in diameter and 90 m in length. A total of 6 μL of the extract was passed down a fused silica capillary column that had a flow rate of 1 mL/min and employed 99.999% helium as the carrier gas. The temperature for the injector and iron source were 250 °C and 280 °C, respectively, while the isothermal temperature was 110 °C for 2 min. The temperature increased at a rate of 10 °C/min to 200 °C, followed by 5 °C/min to 280 °C, and finally 9 min to 280 °C. The mass spectrum was calculated between 50 and 550 *m*/*z*, and the temperatures of the MS quad and the source were kept at 150 and 250 °C, respectively. The total GC-MS analysis process required 36 min. Mass spectra of unidentified peaks were collected and compared to NIST (National Institute of Standards and Technology) databases [44] for the characterization of the compounds.

### 4.4. Primary Culture of Hippocampal Neurons and Extract/Compound Treatment

In accordance with institutional rules and with the approval of the Institutional Animal Care and Use Committee of the College of Medicine at Dongguk University in Korea, all animal care and procedures were performed with the approval certificate number IACUC-2023-06. On the 13th day of pregnancy, rats (Sprague–Dawley) were ordered and kept in a 12 h light–dark cycle with free access to food and fresh water. Isofluorane was used to euthanize the rats, and the fetuses were collected on embryonic developmental day 19. The brains were dissected in order to obtain the hippocampal tissues, and the hippocampal neurons were prepared in the same manner as previously reported [10,45]. In brief, for morphometric analysis of early development, the cells were plated in 24-well plates with poly-DL-lysine (Sigma-Aldrich, St. Louis, MO, USA) coated coverslips and maintained in a specified serum-free neurobasal media supplemented with B27 and incubated at 37 °C with 5% CO_2_ and 95% air. Cells were seeded at a density of about 2 × 10^4^ cells/cm^2^ for morphometric analysis of early development, 6 × 10^4^ cells/cm^2^ for synaptogenesis study, or 6 × 10^4^ cells/cm^2^ for viability assay. For the western blot, the cells were plated at a density of 3 × 10^6^ cells/cm^2^ onto poly-D-lysine (PDL)-coated culture plates. Extracts, compounds, or vehicles were added to the culture media before cells were seeded. 

### 4.5. Neuronal Viability 

After eight days of treatment with extracts or vehicles, trypan blue exclusion assays were used to measure neuronal viability. Neurons were stained with trypan blue for 30 min at 37 °C. After washing with Dulbecco’s phosphate-buffered saline, D-PBS (Invitrogen), neurons were observed under a fluorescence microscope for the quantification of live cells. Due to the reduced membrane permeability of dead neurons, they take up the dye and appear dark blue in hue while dye can’t get into healthy neurons since their membranes are intact. The cell viability was calculated by dividing the number of cells that were labeled by the total number of cells. In three separate studies, cells were counted at random on three coverslips containing 450–500 cells.

### 4.6. Immunofluorescence Staining

Neurons grown on coverslips were at the appropriate days in vitro (DIV) intervals, washed twice with 1× D-PBS, and fixed using a progressive 4% paraformaldehyde/100% methanol fixation procedure. Cells were permeabilized with 0.1% Triton X-100 in PBS for 7 min and after two washes with 1× PBS, neurons were subjected to blocking with 0.4% fish skin gelatin and 2.5% goat serum in 0.2% Tween in PBS. After that, neurons were incubated with primary antibodies that were diluted in antibody diluent (1× PBS, 1% BSA, 0.3% Titon X-100) at 4 °C overnight and finally incubated with secondary antibodies for 2 h at RT and then mounted on slides. The immunostaining was performed by using the following antibodies: mouse anti-tubulin α-subunit (Developmental Studies Hybridoma Bank, University of Iowa, Iowa city, IA, USA, 1:25 dilution), rabbit anti-ankyrin G (Santa Cruz Biotechnology Inc., Delaware Ave, CA, USA, 1:25 dilution), mouse anti-MAP2 (Sigma, Saint Louis, MO, USA, 1:250 dilution), rabbit anti-Tau (AbFrontier, Seoul, Republic of Korea, 1:100 dilution), mouse anti-synaptic vesicle protein 2, SV2 (Developmental Studies Hybridoma Bank, University of Iowa, Iowa city, USA, SV2, 1:25 dilution), chicken anti-postsynaptic density protein 95 (PSD95, a kind gift of Dr. R.S. Walikonis, University of Connecticut, Storrs, CT, USA, 1:1000 dilution), rabbit anti-NR2A and NR2B (1:500) [46], rabbit anti p-GSK-3beta (s9) (Cell Signaling Technology, 1:500 dilution), rabbit anti-phosphorylated p-Erk1/2 (Cell Signaling, Danvers, MA, USA, 1:250 dilution), and secondary antibodies (Alexa Fluor 488-conjugated goat anti-mouse IgG, Alexa Fluor 568-conjugated goat anti-rabbit IgG, Alexa Fluor 568-conjugated goat anti-chicken IgG at a dilution of 1:500 (*v*/*v*), Molecular Probes, Eugene, OR, USA).

### 4.7. Western Blotting

After being scraped and washed with ice-cold 1× D-PBS, the neuronal cells were isolated from six-well cell culture plates. Protein extraction buffer [50 mM Tris–HCl (pH 8.0), NP-40, 0.5% (*w*/*v*), 150 mM NaCl, 1% (*v*/*v*), 1% (*w*/*v*) sodium dodecyl sulfate (SDS) and sodium deoxycholate] and protease inhibitor cocktail (Thermo Scientific, Rockford, IL, USA) were used to lyse the cell pellets following a quick centrifugation step. After incubation for 20 min on ice, the cell lysates were centrifuged in a benchtop microfuge at a speed of 13,000 rpm for 15 min at a temperature of 4 °C. The supernatant was collected. The amount of protein was measured using Advanced Protein Assay Reagent (Cytoskeleton Inc., Denver, CA, USA), with a standard of bovine serum albumin (BSA). Each sample’s cell lysate was boiled for 5 min to dissolve 40 μg of protein in 5× sample buffer [Tris-HCl 60 mM (pH 6.8) with 2% SDS, 14.4 mM—mercaptoethanol, 25% glycerol, and 0.1% bromophenol blue]. The prepared sample was then loaded and run onto an SDS-polyacrylamide gel, and the gel was subsequently blotted onto PVDF (polyvinylidene difluoride) membrane. By incubating the membranes in TBS-T (with 0.05% Tween-20) containing 5% skim milk for one hour at RT, the membranes’ non-specific binding was blocked. The membranes were subjected to the incubation process at 4 °C overnight with primary antibodies specific for postsynaptic density-95 (PSD95, chicken polyclonal UCT-c1; 1:1000), NR2A, and NR2B (rabbit polyclonal; 1:1000), and tubulin α-subunit (mouse monoclonal 12G10; 1:1000) as a loading control. Blots were incubated with secondary antibodies conjugated with horseradish peroxidase (1:10,000; anti-rabbit, anti-chicken IgG; Amersham Biosciences, Buckinghamshire, UK) after being washed with TTBS for 2 h at RT. Enhanced Chemiluminescence Detection Kit, West Save Gold (AbFrontier, Young in Frontier, Seoul, Korea) was used to identify immunoreactive bands. Protein band intensities were quantified using Image J (ver. 1.49) software.

### 4.8. Image Acquisition, Analysis, and Quantification

The cells were observed and imaged with a fluorescence microscope (Leica DFC3000G, Wetzlar, Germany) with a Sony^®^ CCD monochrome sensor P of 1.3 megapixels, and Leica Application SuiteX (LasX Version: 3.7.2.22383) microscope software and processed with Adobe Photoshop 7.0 software. ImageJ (version 1.49) software was used to perform the morphometric analysis (number and total length of primary dendrites, axonal length, axonal and dendritic branching orders, and puncta number for synapse) while the neurite tracer (NIH, Bethesda, MD, USA) Sholl plug-in (http://biology.ucsd.edu/labs/ghosh/software, accessed on 28 December 2022) was used to assess the levels of arborization that were present in both axonal and dendritic trees. We defined axonal and dendritic intersections as the sites at which an axon, a dendrite, or their branch intersected at a specific set of concentric circles, and the branching points were identified between two successive concentric circles using a radial distance of 10 μm between each pair of circles. The Olympus BX53 microscope from Olympus in Tokyo, Japan, with DP74 1/1.2-inch Color CMOS camera and Cellsens software Version: 1.18) was used to acquire the images for the synaptic puncta study.

### 4.9. In Silico Network Pharmacology

#### 4.9.1. ADME/T Analysis of Linalool

The two-dimensional structure of linalool was obtained from the PubChem database (https://pubchem.ncbi.nlm.nih.gov/, accessed on 9 August 2023) and then prepared with the Ligprep2.5 module of Schrodinger Suite (2017-1). After that, QikProp (Schrödinger Release 2017-3: QikProp, Schrödinger, LLC, New York, NY, USA) was used to determine the pharmacokinetic parameters including absorption, distribution, metabolism, and excretion/transport (ADME/T).

#### 4.9.2. Screening of the Common Target Genes


*Target identification and network construction*


To predict the targets of linalool, the comparative toxicogenomic database (CTD) (https://ctdbase.org/, accessed on 9 August 2023), SwissTarget (http://www.swisstargetprediction.ch/, accessed on 9 August 2023), TargetNet (http://targetnet.scbdd.com/, accessed on 9 August 2023), and SEA (https://sea.bkslab.org/, accessed on 9 August 2023) were utilized. The Target-Net dataset was screened using a cutoff score of greater than 0.5, while no cut-off was applied for other databases. Neuronal development (ND)-related targets were collected from The GeneCards database (https://www.genecards.org/, accessed on 9 August 2023). In order to find the common targets, a Venn diagram was generated by crossovers between the compound and ND targets.

A PPI network analysis was carried out using Cytoscape v3.9.0 and STRING 11.0 (https://string-db.org/, accessed on 10 August 2023), which applied linalool-ND common targets that had a total score of greater than 0.4. A cytoscape network analyzer tool was applied in order to find out the topological characteristics of the generated network.

#### 4.9.3. Gene Ontology (GO) and Kyoto Encyclopedia of Genes and Genomes (KEGG) Pathway Enrichment Analyses

GO and KEGG studies were carried out using Shiny GO 0.77 (http://bioinformatics.sdstate.edu/go/, accessed on 11 August 2023). The results were shown by a lollipop chart and “Homo sapiens” were selected as the only species. In the enrichment analysis, the KEGG pathway was utilized to depict the consequence of enriched GO terms for biological processes (BP), cellular components (CC), and molecular functions (MF). After that, KEGG mapper was obtained for the neurotrophin-signaling pathway identifying major genes that might be modulated by the bioactive component of the BGE and WGE.

### 4.10. Data Analysis 

To compare data, we used either the Student’s *t*-test or one-way analysis of variance (ANOVA), followed by Duncan’s multiple comparisons tests (GraphPad Prism v 8.0 Software, San Diego, CA, USA). *p* values < 0.05 were considered statistically significant. All of the results from the experiment were summarized using the standard error of the mean (S.E.M.) of three separate replicates of each experiment.

## 5. Conclusions

The present findings revealed that both fresh and black garlic promoted neurite outgrowth, stimulated axonal maturation and dendritic arborization, and modulated synaptic formation in hippocampal neurons by recruiting both presynaptic and postsynaptic proteins, suggesting that garlic extracts could play an important role in the development of neuronal morphology. The GC-MS-based phytochemical analysis offered several metabolites, including linalool in particular, that potentiated neuritogenesis. Network pharmacology study also showed linalool, BGE, and WGE-mediated neuritogenic effects by the downstream signaling pathways, including Erk1/2 phosphorylation and GSK3β phosphorylation. These findings suggest that garlic and its bioactive metabolite linalool could be further exploited for the development of potential therapeutic agents against neurodegenerative disorders.

## Figures and Tables

**Figure 1 ijms-24-13386-f001:**
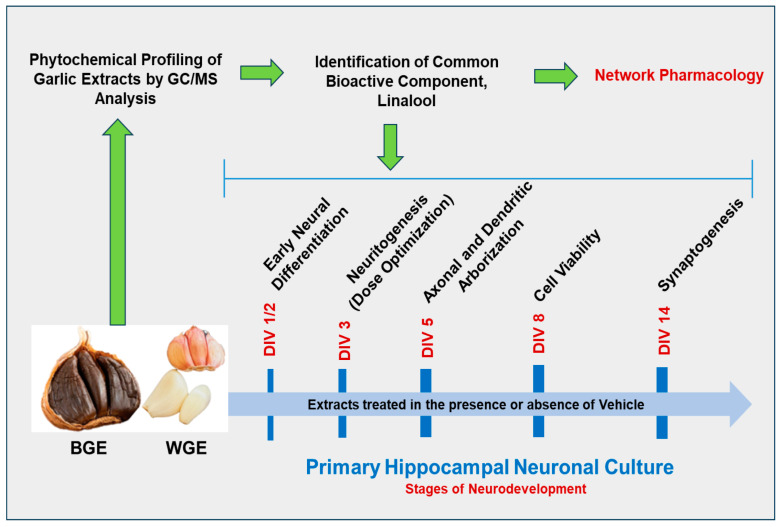
Schematic diagram showing the experimental design conducted in the present study.

**Figure 2 ijms-24-13386-f002:**
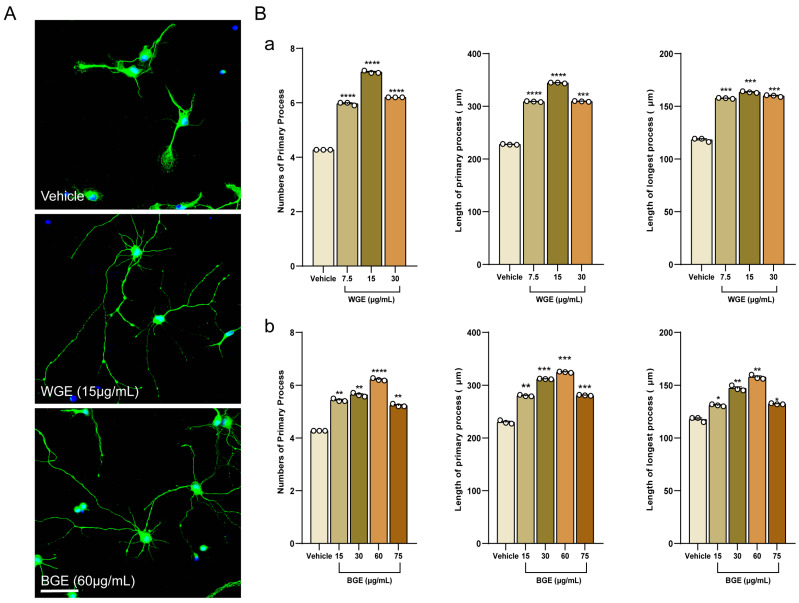
Optimization of WGE and BGE dosages for the neurite development of hippocampus neurons. (**A**) Representative fluorescent images showing neurite outgrowth immuno-stained with α-tubulin (green) and mounted the sample on slide glass with DAPI (blue); scale bar: 50 μm. (**B**) Bar graph showing the statistical analysis of neurite growth including the neurites number, the length of the neurites, and the length of the longest neurites while neurons treated with WGE (**a**) and BGE (**b**). Statistics show as mean ± standard error of the mean (S.E.M.) of three separate experiments (*n* = 3, each includes 10 neurons, one-way ANOVA, * *p* < 0.05, ** *p* < 0.01, *** *p* < 0.001, **** *p* < 0.0001).

**Figure 3 ijms-24-13386-f003:**
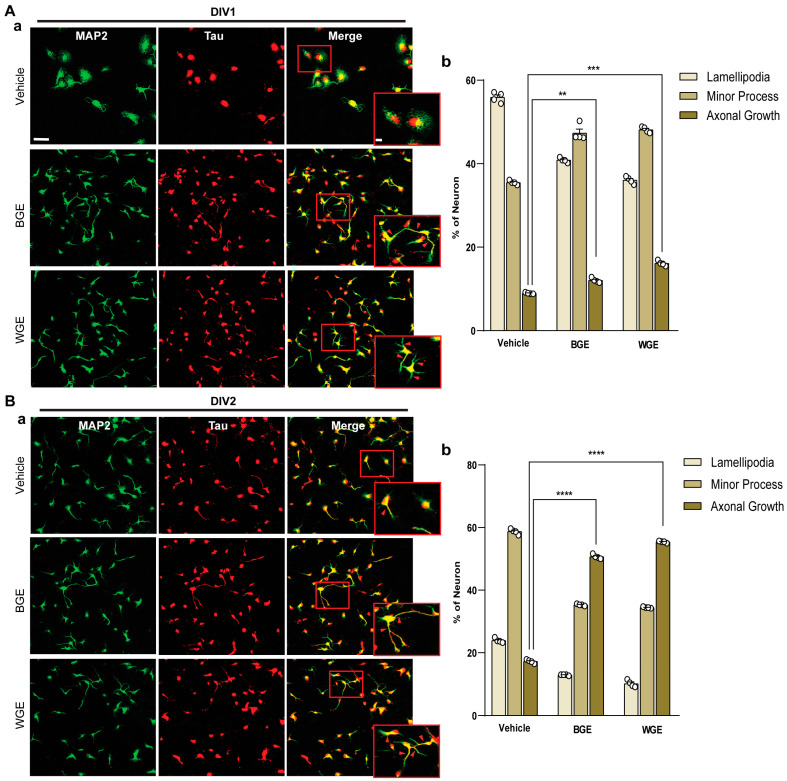
BGE and WGE stimulate early neuronal development. (**A**) Fluorescent images showing the effect of BGE (60 μg/mL) and WGE (15 μg/mL) on early neuronal differentiation after 24 h of plating (DIV1). Here, the major three stages (lamellipodia, minor process, and axonal growth) were marked by red arrowheads (**a**), the percentage of neurons that showed various stages of development after 24 h of incubation (**b**). (**B**) Fluorescent images depicting neuronal polarity after 48 h of plating (DIV2) (**a**), the percentage of neurons at various phases of development at 48 h (**b**). All images in the left panel are of 50 μm scale bar, while all images in the enlarged views are of 5 μm. The values indicated in the bars represent the percentages of between 400 and 500 neurons (*n* = 4) and each value is the mean ± S.E.M. (standard error of the mean, Student’s *t*-tests). ** *p* < 0.01, *** *p* < 0.001, **** *p* < 0.0001).

**Figure 4 ijms-24-13386-f004:**
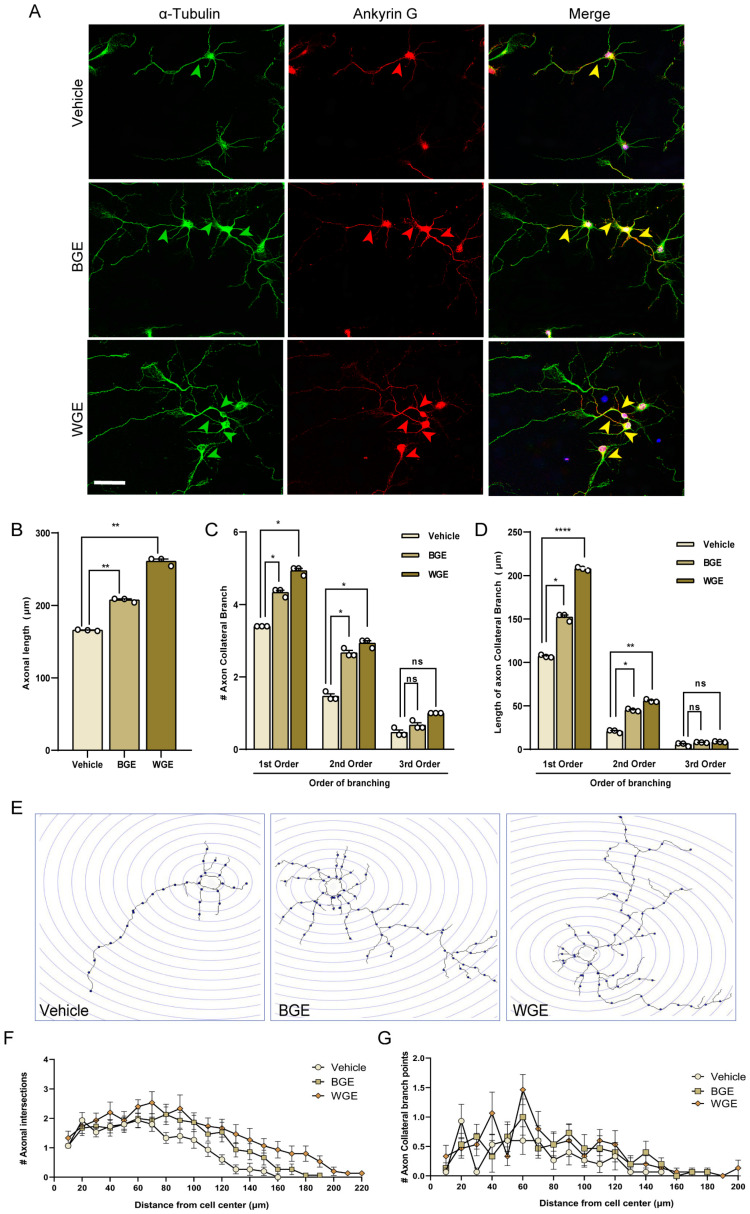
Effect of BGE and WGE on axonal sprouting of hippocampal neurons showed by red (axon) and green (the neuron’s cytoskeleton) arrows. Neurons from the hippocampus were grown for five days under the same conditions as shown in Figure 1, with or without WGE (15 μg/mL) and BGE (60 μg/mL). After being fixed, the neurons were double immune-stained for the axon-specific marker, ankyrin G (red), and for symbolizing the neuron’s cytoskeleton, α-tubulin (green) and mounted the sample on slide glass with DAPI (blue). (**A**) Representative photomicrographs demonstrating the cell morphology; scale bar = 50 μm. (**B**) Morphometric study of axonal arborization: axonal length of WGE and BGE-treated neurons compared to vehicle, (**C**) axon collateral branch, (**D**) length of axon collateral branch. (**E**) A representation of Sholl image of reconstructed neurons showing the axonal sprouting of WGE and BGE-treated neurons compared to vehicle. Sholl analysis indicates (**F**) the number of axon intersections and (**G**) collateral axon branching points. Statistics show as mean ± standard error of the mean (S.E.M.) of three separate experiments (*n* = 3, each includes 10 neurons, Student’s *t*-tests, * *p* < 0.05, ** *p* < 0.01, **** *p* < 0.0001, ns: not significant and numbers are indicated by # symbol).

**Figure 5 ijms-24-13386-f005:**
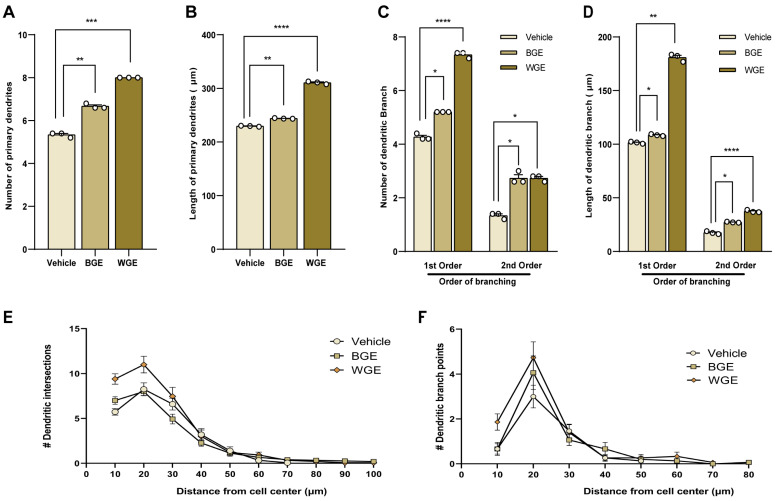
Effect of BGE and WGE on dendritic maturation. The microscopy images that are representative of dendritic morphogenesis are identical to those shown in Figure 4A. (**A**) Morphometric analysis shows statistically significant in the number of primary dendrites, (**B**) the total length of primary dendrites, (**C**) the number, and (**D**) the total length of dendritic branches. (**E**) Sholl analysis for dendritic intersection points and (**F**) dendritic branching points. Statistics show as mean ± standard error of the mean (S.E.M.) of three separate experiments (*n* = 3, each includes 10 neurons, Student’s *t*-tests, * *p* < 0.05, ** *p* < 0.01, *** *p* < 0.001, **** *p* < 0.0001) and numbers are indicated by # symbol.

**Figure 6 ijms-24-13386-f006:**
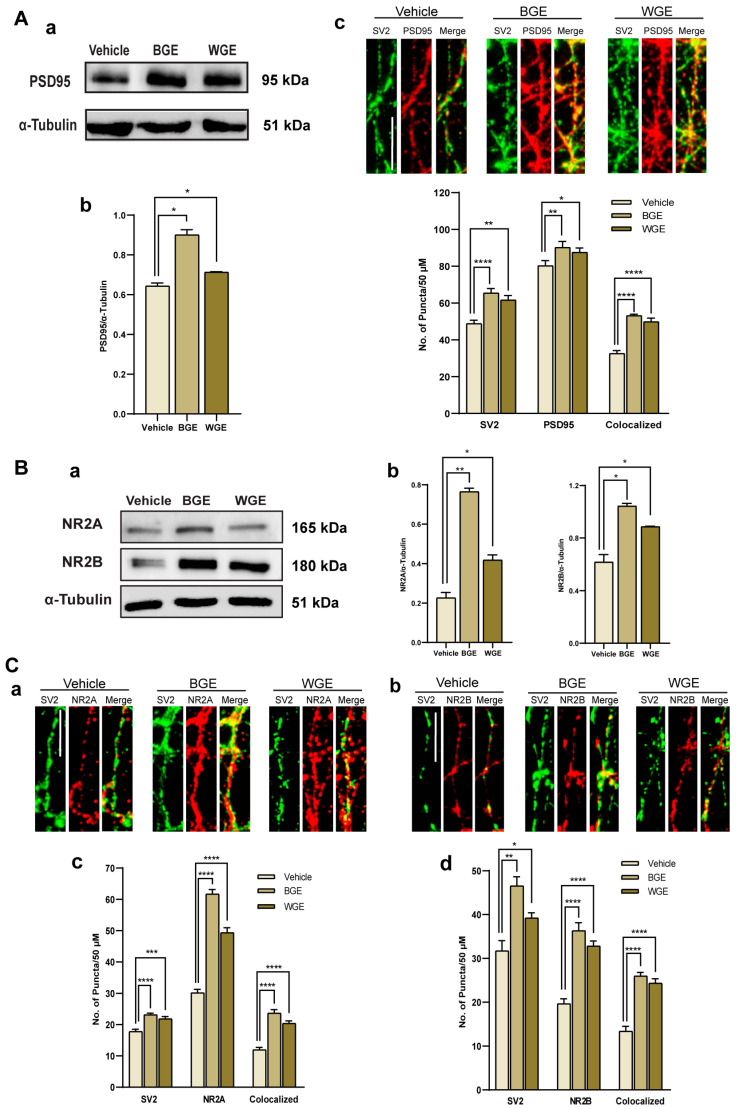
Effect of BGE and WGE on synaptic sprouting. (**A**) Immunoblotting analysis of neuronal lysates from DIV16 confirmed the presence of the proteins PSD95 compared with α-tubulin (**a**), and a comparison of mean intensity revealed a statistically significant difference in PSD95 expression between BGE and WGE-treated neurons (**b**), fluorescence images showing the pre-synapse and post-synapse puncta are indicated by co-staining with SV2 (green) and PSD95 (red), respectively. Co-localization (yellow puncta) indicates synapse formation; scale bar: 2 μm, and graph depicting the number of SV2, PSD95, and co-localized (synapse) puncta that are found per 50 μm segment in neurons that were either treated with or without WGE and BGE (**c**). (**B**) Immunoblotting analysis of neuronal lysates from DIV16 confirmed the presence of the proteins GluN2A (NR2A) and GluN2B (NR2B) compared with α-tubulin as a loading control (**a**), and a comparison of mean intensity revealed a statistically significant difference in NMDA receptor expression between BGE and WGE-treated neurons (**b**). (**C**) Fluorescence images showing the immune-staining of SV2 (green) with NR2A (red) (**a**), NR2B (red) (**b**), and co-localized (synapse) puncta that are found per 50 μm segment in neurons that were either treated with or without WGE and BGE, scale bar: 2 μm; the bar graph depicting the numbers of SV2, NR2A puncta (**c**), SV2, NR2B puncta, (**d**) and co-localized (synapse) puncta that are found per 50 μm segment in neurons that were either treated with or without WGE and BGE. Statistics show as mean ± standard error of the mean (S.E.M.) of three separate experiments (*n* = 3, each includes 10 neurons, Student’s *t*-tests, * *p* < 0.05, ** *p* < 0.01, *** *p* < 0.001, **** *p* < 0.0001).

**Figure 7 ijms-24-13386-f007:**
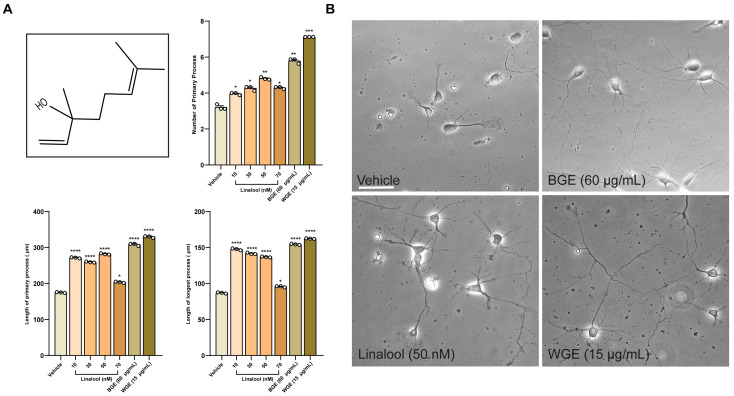
Linalool promotes neurite outgrowth. (**A**) Structure of linalool and morphological data showing the numbers of the primary process, the total length of the primary process, and the length of the longest process. Statistics show as mean ± standard error of the mean (S.E.M.) of three separate experiments (*n* = 3, each includes 10 neurons, ANOVA, * *p* < 0.05, ** *p* < 0.01, *** *p* < 0.001, **** *p* < 0.0001). (**B**) Representative bright field images showing the effect of different doses of linalool in comparison to the vehicle; scale bar, 50 µm.

**Figure 8 ijms-24-13386-f008:**
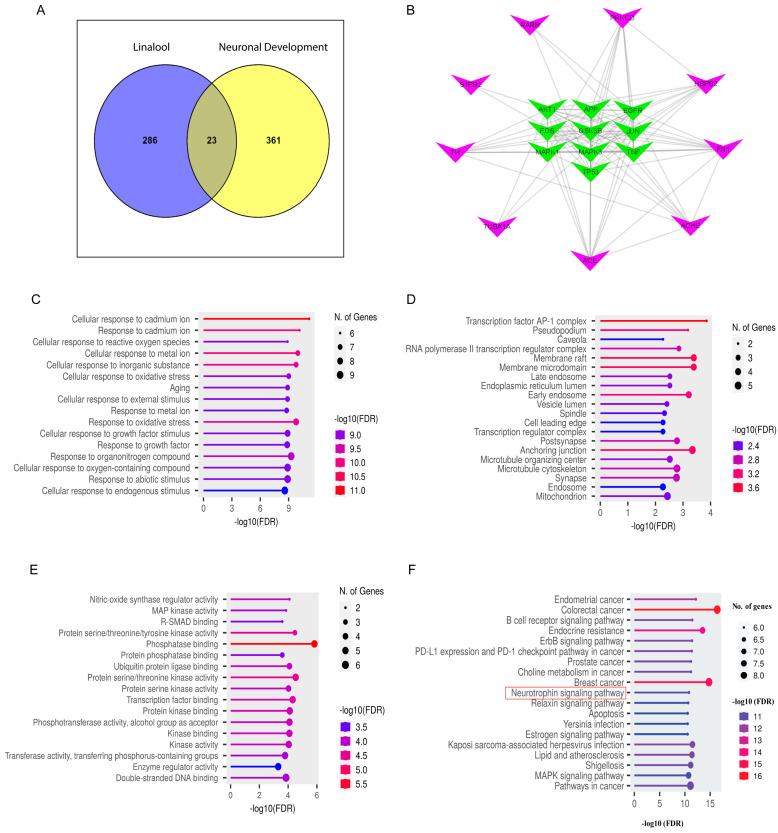
Identifying potential targets of active compound linalool for neuronal development. (**A**) Venn diagram of the common target of linalool-ND genes. (**B**) Protein-protein interaction network for common target genes where green color nodes denote the top 10 hub genes with the greatest degrees’ score. (**C**) GO enrichment analysis chart for the term in the biological function (BP), (**D**) cellular components (CC), and (**E**) molecular function (MF) category. (**F**) KEGG pathway enrichment analysis for the top 20 significant cellular pathways.

## Data Availability

The data will be provided upon reasonable request.

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
