# Peer review of "Differential Effects of the Processed and Unprocessed Garlic (*Allium sativum* L.) Ethanol Extracts on Neuritogenesis and Synaptogenesis in Rat Primary Hippocampal Neurons"

_ijms, 2023, doi:10.3390/ijms241713386_

Round 1
Reviewer 1 Report
The manuscript “Differential Effects of the Processed and Unprocessed Garlic (Allium Sativum L.) Ethanol Extracts on Neuritogenesis and Synaptogenesis in Rat Primary Hippocampal Neurons” by Yeasmin Akter Munni et al. The manuscript is written in standard English; however, it has several grammatical and typographical errors. After thoroughly reviewing it, I feel the manuscript needs substantial revision.
Comments:
1. In the abstract section, I suggest better rewriting the conclusion.
2. I will suggest citing the latest literature in the introduction section and rewriting it properly.
3. I will suggest adding experimental design as a Schematic diagram
4. The authors should carefully check grammatical errors.
5. I will suggest explaining the discussion section properly.
6. I will suggest adding the molecular weight of the specific protein in the western blot.
7. Authors have performed only protein expression experiments to prove their hypothesis. I will suggest that authors can add a few genes related to their hypothesis that can strengthen the data
8. Generally, proteins are isolated by centrifugation for 15-20 minutes. However, the authors have done it for 5 minutes…must be cited.
9. In the western blot blocking solution, authors have used 0.05% Tween-20. Generally, 0.01% is used for blocking the solution. Was there any specific reason? Either answer or must be cited.
10. Generally, secondary antibodies dilution for western blot is 1: 5000-10000 recommended by the manufacturers, however, authors have used 1;1000
Author Response
The manuscript “Differential Effects of the Processed and Unprocessed Garlic (Allium Sativum L.) Ethanol Extracts on Neuritogenesis and Synaptogenesis in Rat Primary Hippocampal Neurons” by Yeasmin Akter Munni et al. The manuscript is written in standard English; however, it has several grammatical and typographical errors. After thoroughly reviewing it, I feel the manuscript needs substantial revision.
Authors response:
The authors would like to appreciate the valuable time and critical criticism of the reviewer. We have responded to each comment point by point.
Comments:
- In the abstract section, I suggest better rewriting the conclusion.
Response:
Thank you for the suggestion. The conclusion in the abstract section has been revised as lines 41 to 43.
“Overall, these findings provide information on the molecular mechanism of processed and unprocessed garlic for neuronal growth, survival, and memory function which may have the potential for the prevention of several neurological disorders.”
- I will suggest citing the latest literature in the introduction section and rewriting it properly.
Response:
We have revised and cited some latest literature in the introduction section. The latest references are highlighted in green color.
- I will suggest adding experimental design as a Schematic diagram
Response:
Thank you for the suggestion. We have prepared one experimental design as a Schematic diagram and mentioned it in Figure 1.
- The authors should carefully check grammatical errors.
Response:
We have checked all grammatical errors.
- I will suggest explaining the discussion section properly.
Response:
We have made some edits to the discussion and aimed to explain the whole thing accurately as follows:
“Overall, these findings provide information on the molecular mechanism of processed and unprocessed garlic for the neuronal growth, survival, and memory function which may have the potential for the prevention of several neurological disorders” Lines 387 to 392
“Numerous signaling pathways and proteins are responsible to promote neuronal differentiation and development. Using the network pharmacology approach, we demonstrated the top ten genes with high degree scores from PPI identifying several genes implicated in neurotrophic activity including GSK3β, c-Jun, Erk1/2, Akt, and P53 by KEGG pathway enrichment analysis. p- GSK3β and p- Erk1/2 were found to be regulated in the linalool, BGE, and WGE-treated neurons. For instance, emerging evidence suggests that GSK3β is a significant modulator of neuronal development such as neuronal polarization, axon development, and axonal sprouting [37] while GSK3β activity was decreased by phosphorylation at its Ser-9 [38, 39]. Both BGE and WGE and the bioactive compound, linalool-treated neurons reduced the expression of p- GSK3β compared to the vehicle indicating the GSK3β-promoting activity in neuronal development. In addition, BGE, WGE, and their active metabolite, linalool increased the expression of p-Erk1/2. MEK1/2 (MAPK) is an upstream activator kinase that is known to activate Erk 1/2 by phosphorylating threonine and tyrosine residues which in turn regulates the transcription of several genes involved in neurite outgrowth and axonal, dendritic sprouting, synaptogenesis, and memory formation [40-42]. Therefore, BGE and WGE might upregulate Erk1/2 and GSK3β signaling pathways to promote neuritogenesis and synaptogenesis.
Overall the combined studies from in vitro and network pharmacology approach suggest the mechanistic understanding of processed and unprocessed garlic on neuronal differentiation and synaptogenesis” Lines 438 to 457
- I will suggest adding the molecular weight of the specific protein in the western blot.
Response:
We have added the molecular weight of the specific protein in the western blot (Figure 6).
- Authors have performed only protein expression experiments to prove their hypothesis. I will suggest that authors can add a few genes related to their hypothesis that can strengthen the data
Response:
Thank you for the suggestion. Using network pharmacology, we were able to identify several genes implicated in neurotrophic activity (Figure 8, Supplementary figure S5), including GSK3β (glycogen synthase kinase 3), c-Jun (Jun proto-oncogene AP-1 transcription factor subunit), ERK1/2 (extracellular signal-regulated protein kinase), Akt (AKT serine/threonine kinase 1), and P53 (tumor protein p53). The gene expression of p-GSK3β and pERK1/2 were validated by the immunocytochemistry in supplementary figure S6. The description of network pharmacology and ICC from lines 330 to 374 are as follows:
“2.9. Network pharmacology analysis
To find out the toxicity and bioavailability of linalool, we performed ADME/T analysis. The #stars score was less than 4 (ranging from 0 – 5) while the brain/blood partition coefficient (QPlogBB) was found at 0.015 (–3.0 – 1.2, recommended range). Linalool satisfied the rules of five by maintaining the mol_MW < 500, donor HB ≤5, accptHB ≤10 and QPlogPo/w < 5 (Supplementary File 2).
A total of 309 genes were identified as linalool’s potential targets. Likewise, by using the Gene Cards dataset, 384 target genes associated with neuronal development (ND) were collected (Supplementary File 2). An overlap of 23 target genes was found as linalool-ND common targets which was shown in figure 8A through a Venn diagram. The PPI network was generated with Cytoscape software (Figure 8B) following String databases where 10 hub genes were represented by green nodes.
GO enrichment analysis showed that the biological process (BP) term was mostly involved in response to oxidative stress, cellular response to growth factor stimulus, and response to growth factor (Figure 8C). Postsynapse, synapse, microtubule skeleton, cell leading edge, and microtubule organizing center were seen in the cellular component (CC) term (Figure 8D). On the other hand, the molecular function (MF) term was associated with the phosphatase binding, protein serine/threonine kinase activity, transcription factor binding and enzyme regulator activity (Figure 8E). The top 20 most important signaling pathways were selected for KEGG enrichment pathway analysis (Figure 8F). Among the various cell signaling pathways, the neurotrophin signaling pathway, and MAPK signaling pathway were observed that are essential for neuronal survival, neurite formation, and development, which includes the formation of synapses and synaptic plasticity. Additionally, the KEGG pathway mapper (Figure S5, Supplementary File 1) was also analyzed and it showed five major targets involved in neuronal development. These targets are GSK3β (glycogen synthase kinase 3), c-Jun (Jun proto-oncogene AP-1 transcription factor subunit), Erk1/2 (extracellular signal-regulated protein kinase), Akt (AKT serine/threonine kinase 1), and P53 (tumor protein p53) which might be potentially modulated by the bioactive component of BGE and WGE.
2.10. Validation of target genes by immunocytochemistry
Using ICC, we evaluated the expression of the genes during neural differentiation (p-Erk1/2, p-GSK3β) to confirm the modulation of distinct genes by linalool as found in the network pharmacology (Figure S6 A-a, B-a, Supplementary File 1). The relative fluorescence intensity of Erk1/2 were significantly increased (Figure S6A-b, Supplementary File 1) while p-GSK3β expression was decreased in linalool, BGE and WGE-treated hippocampal neurons (Figure S6 B-b, Supplementary File 1) , indicating the differential role of linalool and the parent extracts in neurite development in primary hippocampal neurons”
- Generally, proteins are isolated by centrifugation for 15-20 minutes. However, the authors have done it for 5 minutes…must be cited.
Response:
We are genuinely appreciating your attention to this point and we sincerely apologize for the error. We always isolate the protein by centrifugation for 15 minutes. The time has been edited from 5 minutes to 15 minutes as follows:
“After incubation for twenty minutes on ice, the cell lysates were centrifuged in a benchtop microfuge at a speed of 13000 rpm for 15 minutes at a temperature of 40C” Lines 572 to 574.
- In the western blot blocking solution, authors have used 0.05% Tween-20. Generally, 0.01% is used for blocking the solution. Was there any specific reason? Either answer or must be cited.
Response:
Previously, our lab used to prepare the blocking solution with 0.01% Tween-20 but we had experienced several nonspecific bands in the membrane. To reduce the nonspecific binding sites, we have increased the concentration of Tween-20 to 0.05%.
"By incubating the membranes in TBS-T (with 0.05% Tween-20) containing 5% skim milk for one hour at RT, the membranes' non-specific binding was blocked” Line 581
- Generally, secondary antibodies dilution for western blot is 1: 5000-10000 recommended by the manufacturers, however, authors have used 1;1000
Response:
The secondary antibodies dilution for western blot would be 1:10000. It was a mistake in typing. We have corrected it in the method section (4.7. western blot).
“Blots were incubated with secondary antibodies conjugated with horseradish peroxidase (1:10000; anti-rabbit, anti-chicken IgG; Amersham Biosciences, Buckinghamshire, UK) after being washed with TTBS for 2 hours at RT” Lines 586 to 588.
Reviewer 2 Report
Review of the manuscript entitled: Differential Effects of the Processed and Unprocessed Garlic (Allium Sativum L.) Ethanol Extracts on Neuritogenesis and Synaptogenesis in Rat Primary Hippocampal Neurons. In my opinion the manuscript is well prepared. The manuscript is interesting and touches on a very important topic but some corrections need to be made.
In introduction clear aim of the manuscript should be added e.g. "The aim of the present study was to ...". Moreover introduction must be ended with the aim of the manuscript.
The description of the results is fundamentally wrong! It is unacceptable to interpret the results, discuss them and references in the “Results” chapter. The Authors do not understand the difference between the results and discussion chapters? The entire results section must be signed, e.g. lines 84-85, 94-96 should be deleted or transfer to discussion, please correct the rest. If you want to discuss the results immediately, delete the “Discussion” section.
Was the study approved by the bioethics committee?
Author Response
The authors would like to appreciate the valuable time and critical criticism of the reviewer. We have responded to each comment point by point.
Review of the manuscript entitled: Differential Effects of the Processed and Unprocessed Garlic (Allium Sativum L.) Ethanol Extracts on Neuritogenesis and Synaptogenesis in Rat Primary Hippocampal Neurons. In my opinion the manuscript is well prepared. The manuscript is interesting and touches on a very important topic but some corrections need to be made.
- In introduction clear aim of the manuscript should be added e.g. "The aim of the present study was to ...". Moreover introduction must be ended with the aim of the manuscript.
Response:
Thank you very much for taking the time to review our manuscript. We have discussed the purpose of our present study from lines “60 to 63” as follows:
“Focusing on the neurotrophic effects of the bioactive components, the aim of this present study is to investigate the progression of neurons from the nascent neuronal differentiation stage to synapse formation of ethanol extract of unprocessed and processed A. sativum L.”
We have also modified the end of the introduction from lines 88 to 92 as follows
“Moreover, the network pharmacology technique was subsequently implemented to gain mechanistic insight into the bioactive component, and the potential target was confirmed by immunocytochemistry. Together, this combined study of in vitro and network pharmacology showed that BGE and WGE promote neuronal differentiation, axodendritic outgrowth, and synaptic development.”
- The description of the results is fundamentally wrong! It is unacceptable to interpret the results, discuss them and references in the “Results” chapter. The Authors do not understand the difference between the results and discussion chapters? The entire results section must be signed, e.g. lines 84-85, and 94-96 should be deleted or transfer to discussion, please correct the rest. If you want to discuss the results immediately, delete the “Discussion” section.
Response:
Thank you for the suggestion. We have deleted the explanation of the findings, including lines 84 and 85 (now 95 to 97), as well as the rest of the results section.
- Was the study approved by the bioethics committee?
Response:
Yes. This study was approved by the bioethics committee and it was mentioned in the method section from lines 514 to 515 as follows:
"all animal care and procedures were performed with the approval certificate number IACUC-2023-06”
Round 2
Reviewer 1 Report
The manuscript has improved, however, figure 6 (B) molecular weight of the protein is still missing.